# Using symptom-based case predictions to identify host genetic factors that contribute to COVID-19 susceptibility

Irene V. van Blokland[1,2☯], Pauline Lanting[1☯], Anil P. S. Ori[1,3☯], Judith M. Vonk[4☯], Robert C. A. Warmerdam[1☯], Johanna C. Herkert[1], Floranne Boulogne[1], Annique Claringbould[1,5], Esteban A. Lopera-Maya[1], Meike Bartels[6,7], Jouke-Jan Hottenga[6], Andrea Ganna[8], Juha Karjalainen[8,9,10], Lifelines COVID-19 cohort study[¶], The COVID-19 Host Genetics Initiative[¶], Caroline Hayward[11], Chloe Fawns-Ritchie[12], Archie Campbell[13], David Porteous[13], Elizabeth T. Cirulli[14], Kelly M. Schiabor Barrett[14], Stephen Riffle[14], Alexandre Bolze[14], Simon White[14], Francisco Tanudjaja[14], Xueqing Wang[14], Jimmy M. Ramirez, III[14], Yan Wei Lim[14], James T. Lu[14], Nicole L. Washington[14], Eco J. C. de Geus[6,7], Patrick Deelen[1,15], H. Marike Boezen[4‡], Lude H. Franke[1‡]*

1 Department of Genetics, University of Groningen, University Medical Center Groningen, Groningen, The Netherlands, 2 Department of Cardiology, University of Groningen, University Medical Center Groningen, Groningen, The Netherlands, 3 Department of Psychiatry, University of Groningen, University Medical Center Groningen, Groningen, The Netherlands, 4 Department of Epidemiology, University of Groningen, University Medical Center Groningen, Groningen, The Netherlands, 5 Structural Computational Biology unit, EMBL, Heidelberg, Germany, 6 Department of Biological Psychology, FGB, Vrije Universiteit Amsterdam, Amsterdam, The Netherlands, 7 Amsterdam Public Health research institute, Amsterdam UMC, Amsterdam, The Netherlands, 8 Institute for Molecular Medicine Finland, University of Helsinki, Helsinki, Finland, 9 Broad Institute of MIT and Harvard, Cambridge, MA, United States of America, 10 Analytic and Translational Genetics Unit (ATGU), Massachusetts General Hospital, Boston, MA, United States of America, 11 MRC Human Genetics Unit, Institute of Genetics and Molecular Medicine, University of Edinburgh, Edinburgh, United Kingdom, 12 Department of Psychology, University of Edinburgh, Edinburgh, United Kingdom, 13 Medical Genetics Section, Centre for Genomic and Experimental Medicine, Institute of Genetics and Molecular Medicine, University of Edinburgh, Edinburgh, United Kingdom, 14 Helix OpCo LLC, San Mateo, California, United States of America, 15 Department of Genetics, University Medical Centre Utrecht, Utrecht, The Netherlands

☯ These authors contributed equally to this work.
‡ These authors are senior authorship on this work.
¶ Membership of the Lifelines COVID-19 cohort study and The COVID-19 Host Genetics Initiative is listed in the Acknowledgments.
* l.h.franke@umcg.nl

**Data Availability Statement:** Due to legal restrictions on sharing the de-identified data, individual level data can not be shared on a public repository. However, data is available upon request

## Abstract

Epidemiological and genetic studies on COVID-19 are currently hindered by inconsistent and limited testing policies to confirm SARS-CoV-2 infection. Recently, it was shown that it is possible to predict COVID-19 cases using cross-sectional self-reported disease-related symptoms. Here, we demonstrate that this COVID-19 prediction model has reasonable and consistent performance across multiple independent cohorts and that our attempt to improve upon this model did not result in improved predictions. Using the existing COVID-19 prediction model, we then conducted a GWAS on the predicted phenotype using a total of 1,865 predicted cases and 29,174 controls. While we did not find any common, large-effect variants that reached genome-wide significance, we do observe suggestive genetic

via the research office of each individual cohort. For contact and access, see information below. All GWAS summary statistics can be downloaded from the C19HG website https://www.covid19hg.org/results/. Contact information individual cohorts: GS:SFHS cohort is available to researchers in the UK and to international collaborators through application to the GS Access Committee. GS operates a managed data access process including an online application form (http://www.gsaccess.org/) and proposals are reviewed by the GS Access Committee. The Helix data were collected under IRB Protocol #20170748. Data are available on reasonable request, which should be directed to research@helix.com The Lifelines data analysed in this study were obtained from the Lifelines biobank, under project application number ov20_0554. Requests to access this dataset should be directed to Lifelines Research Office (research@lifelines.nl). Requests to access the NTR dataset should be directed to ntr.datamanagement.fgb@vu.nl.

**Funding:** Generation Scotland received core support from the Chief Scientist Office of the Scottish Government Health Directorates [CZD/16/6] and the Scottish Funding Council [HR03006] and is currently supported by the Wellcome Trust [216767/Z/19/Z]. Genotyping of the GS:SFHS samples was carried out by the Genetics Core Laboratory at the Edinburgh Clinical Research Facility, University of Edinburgh, Scotland and was funded by the Medical Research Council UK and the Wellcome Trust (Wellcome Trust Strategic Award "STratifying Resilience and Depression Longitudinally" (STRADL) Reference 104036/Z/14/Z). Recruitment to this study was facilitated by SHARE - the Scottish Health Research Register and Biobank. SHARE is supported by NHS Research Scotland, the Universities of Scotland and the Chief Scientist Office of the Scottish Government. C.H. is supported by an MRC University Unit Programme Grant MC_UU_00007/10 (QTL in Health and Disease). Salaries of the authors ETC, KMSB, SR, AB, SW, FT, XW, JMR, YWL, JTL, and NLW was supported by Helix OpCo, LLC. The funding organization did not play a role in the study design, data analysis, decision to publish, or preparation of the manuscript for commercial purposes and only provided financial support in the form of authors' salaries and/or research materials. The Lifelines Biobank initiative has been made possible by funding from the Dutch Ministry of Health, Welfare and Sport; the Dutch Ministry of Economic Affairs; the University Medical Center Groningen (UMCG the Netherlands); the University of Groningen and the Northern Provinces of the

associations at two SNPs (rs11844522, p = 1.9x10-7; rs5798227, p = 2.2x10-7). Explorative analyses furthermore suggest that genetic variants associated with other viral infectious diseases do not overlap with COVID-19 susceptibility and that severity of COVID-19 may have a different genetic architecture compared to COVID-19 susceptibility. This study represents a first effort that uses a symptom-based predicted phenotype as a proxy for COVID-19 in our pursuit of understanding the genetic susceptibility of the disease. We conclude that the inclusion of symptom-based predicted cases could be a useful strategy in a scenario of limited testing, either during the current COVID-19 pandemic or any future viral outbreak.

## Introduction

The Coronavirus Disease 2019 (COVID-19) caused by Severe Acute Respiratory Syndrome Coronavirus-2 (SARS-CoV-2) has rapidly spread across the globe, posing a large burden on individuals, healthcare systems, and societies as a whole. At the time of writing, more than 55 million infections and 1,300,000 deaths have been reported worldwide [1]. The symptoms and disease severity of COVID-19 vary [2], ranging from asymptomatic or nonspecific symptoms to severe illness with hospital admission and death. While the scientific community is rapidly gaining more understanding of the pathophysiology of COVID-19 [3, 4], many questions remain about the etiology of the disease and what factors are driving the interindividual variability in pathophysiology.

It is known that individual genetic differences in the human host contribute to immune function and response to common infectious agents [5, 6]. Genome-wide association studies (GWAS) have, for example, identified susceptibility loci for multiple common infections [7]. The identification of genetic factors can lead to new insights into disease mechanisms and help improve vaccination strategies by optimizing vaccine-induced protection. For this reason, the COVID-19 host genetics consortium (C19HG) was established to discover and study the human genetic variants that modulate the susceptibility of developing COVID-19 symptoms and COVID-19 severity [8]. However, the magnitude of the pandemic, limited testing capacity and inconsistent testing policies have likely resulted in an underrepresentation of the number of true cases. Using only confirmed cases reduces the power of any GWAS to detect associations and may be a source of bias.

Recently, a model was published that predicts the potential presence of COVID-19 based on self-reported disease-related symptoms, which we will refer to as the Menni COVID-19 prediction model [9]. We investigated if potential COVID-19 predicted based on symptoms can help accelerate the search for host genetic factors that contribute to the susceptibility of developing COVID-19 symptoms, which we will refer to as COVID-19 susceptibility, and the heterogeneity of COVID-19 severity. First, we confirmed that the Menni COVID-19 model can identify cases with laboratory confirmed SARS-CoV-2 infection in three independent cohorts. As existing COVID-19 prediction models used features not available in our cohorts, we generated a COVID-19 prediction model optimized to the phenotypes described in our cohorts. Second, as part of the C19HG consortium, we performed genetic analyses on predicted COVID-19 (1,865 cases and 29,174 controls, Fig 1) to search for host genetic factors that contribute to COVID-19 susceptibility and explored possible downstream effects of the loci identified. To assess the validity of the predicted COVID-19 phenotype, we compared these results to the GWAS meta-analyses results based on confirmed COVID-19. We also compared our findings to previously reported genetic associations with several viral infectious

Netherlands. The generation and management of GWAS genotype data for the Lifelines Cohort Study was supported by the UMCG Genetics Lifelines Initiative (UGLI). Lifelines COVID-19 data collection was supported by the Netherlands Organization for Scientific Research (NWO): NWO Spinoza Prize (SPI 92-266 to C.W.). L.F. is supported by an NWO Corona Fast-Track grant (440.20.001), an Oncode Senior Investigator grant, a grant from the European Research Council (ERC Starting Grant agreement number 637640 ImmRisk) and an NWO VIDI grant (917.14.374) NTR Covid-19 data collection and data management was supported by NWO and Netherlands Organisation for Health Research and Development (ZonMW) grants 440.20.022 and 480-15-001/674.

**Competing interests:** The authors have read the journal's policy and have the following competing interests: ETC, KMSB, SR, AB, SW, FT, XW, JMR, YWL, JTL, and NLW are employees of Helix OpCo, LLC, which is a provider of COVID-19 testing services. There are no patents, products in development or marketed products associated with this research to declare. This does not alter our adherence to PLOS ONE policies on sharing data and materials.

diseases to look for genetic factors shared between COVID-19 susceptibility and other viral infectious diseases.

# Materials and methods

## Data collection and preparation

Four separate cohorts contributed data to the presented analysis.

The Lifelines COVID-19 cohort consists of individuals from the Lifelines population cohort and the Lifelines NEXT birth cohort in the Northern part of the Netherlands [10]. Within the Lifelines COVID-19 cohort, questionnaires were sent out to participants over the age of 18 years via email on a weekly basis starting March 30, 2020. Items about COVID-19 infection and perceived symptoms, drug use, mental health and vaccination status were questioned weekly. A comprehensive cohort description has been described previously [11].

The Helix cohort consists of individuals from the Helix DNA Discovery Project, an unselected population of adults from across the United States [12]. COVID-19 questionnaires were emailed to participants in April and May of 2020. The questionnaire format was based on example surveys and suggested symptoms and pertinent information compiled by the C19HG [13].

The Netherlands Twin Register (NTR) consists of members of twin families that had been registered as willing to participate in survey, biobank and experimental research. NTR participants aged 16 years or older (range 16–95) received an online questionnaire at the end of April (wave 1) or the middle of May (wave 2). The questionnaire was modelled on the Lifelines survey and contained items about COVID-19 testing, diagnosis and treatment of COVID-19, perceived flu-like symptoms, drug use, past and present chronic diseases, household composition, work setting and the impact of the corona crisis on their mental health and lifestyle behaviours.

The Generation Scotland cohort consists of individuals over the age of 18 from the Generation Scotland biobank. The Covid Life survey was initially launched on April 17, 2020 with a few hundred individuals to make sure the survey process was working well. The following week, the survey was rolled out by email and letter to all of the current Generation Scotland volunteers. Volunteers were asked questions about the impact the pandemic had on their life and included questions on education, mental health, wellbeing and more.

## Ethics statement

Generation Scotland received ethical permission for the creation of the GS:SFHS study (05/S1401/89 Tayside Committee on Medical Research Ethics A). Generic Research Tissue Bank approval has been granted for use of the resource. (20/ES/0021 East of Scotland Research Ethics Service). All participants signed an informed consent form prior to enrolment.

Helix data were collected under Western IRB Protocol #20170748. All participants signed an informed consent form prior to enrolment.

The Lifelines study was approved by the ethics committee of the University Medical Center Groningen, document number METc2007/152. All participants signed an informed consent form prior to enrolment.

The NTR study was approved by the ethics committee of the Faculty of Behavioural and Movement Sciences, Vrije Universiteit Amsterdam, reference: VCWE-2020-083. All participants signed an informed consent form prior to enrolment.

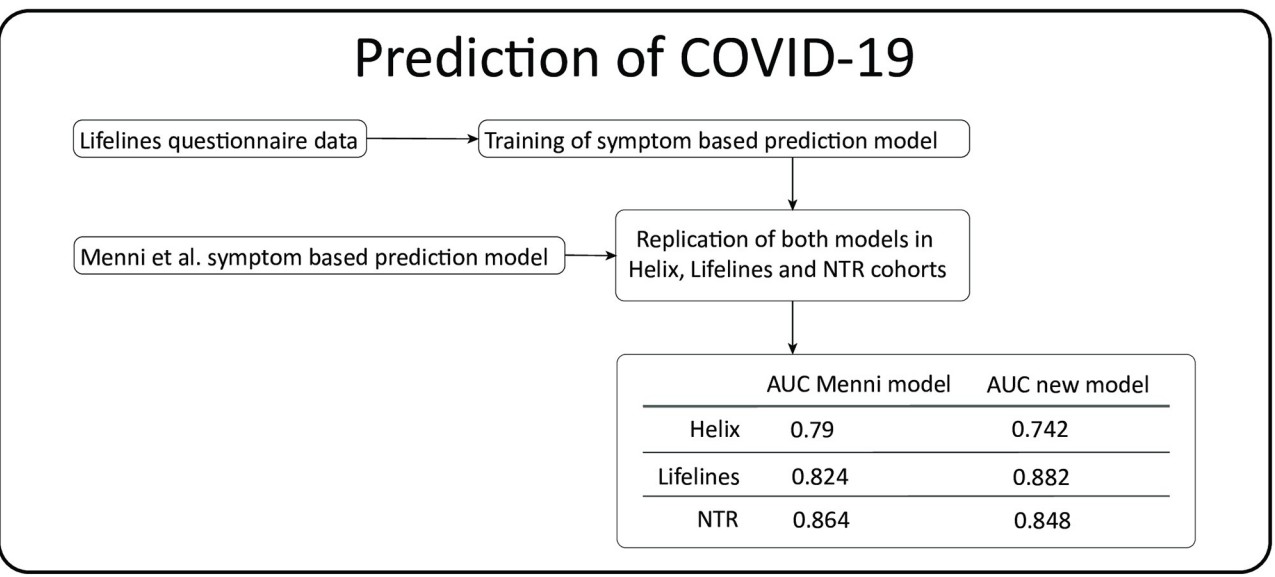

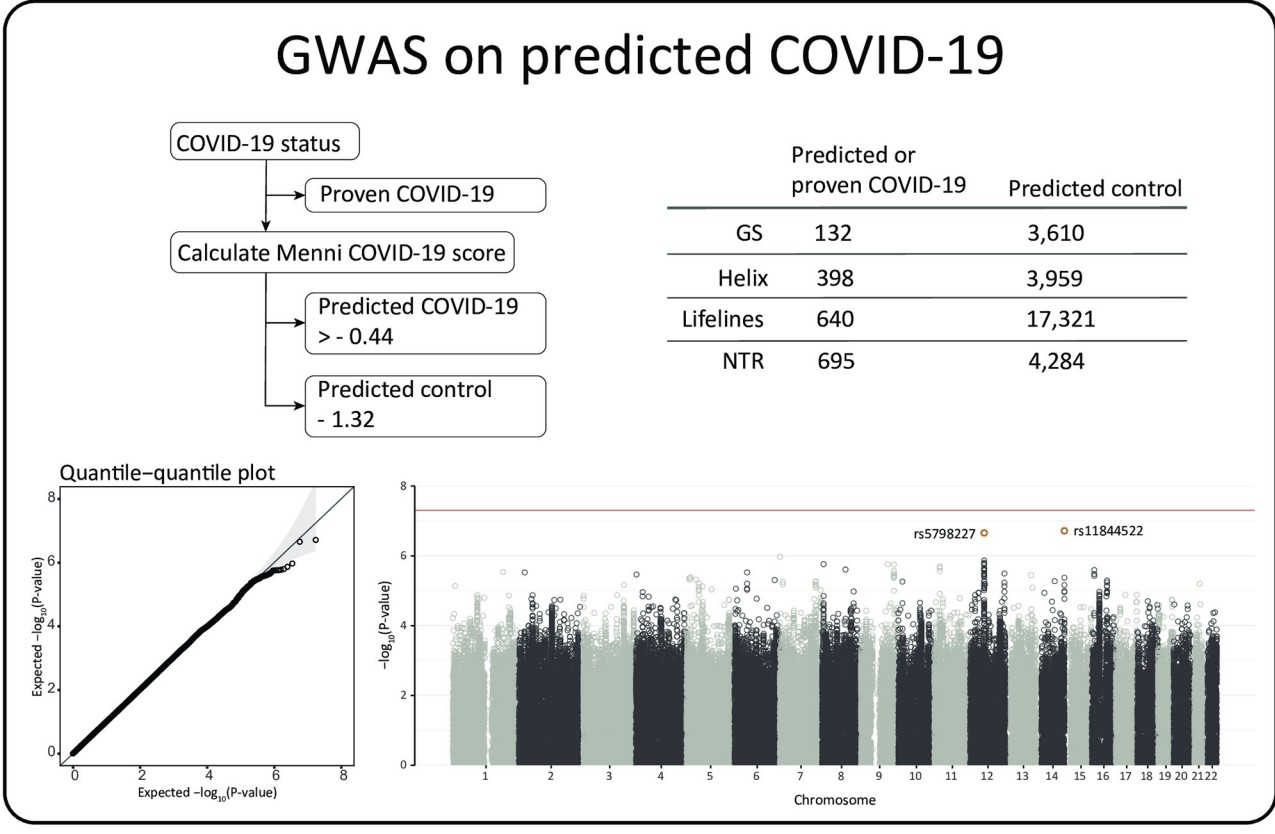

**Fig 1. Overview of the main analysis.**

### The Menni COVID-19 prediction model

The predictive properties of the Menni COVID-19 prediction model were investigated in the Helix, Lifelines, and NTR cohorts separately. The Generation Scotland cohort could not be

included in this analysis since self-reported SARS-CoV-2 reverse-transcription PCR (RT-PCR) test outcomes were not available. In the three cohorts with RT-PCR test outcomes available, symptoms that best captured the included symptoms in the Menni COVID-19 prediction model were selected and non-binary answers (5- or 7-point Likert scale answers) were recoded into binary responses using cut-off values as presented in S1 Table. We applied the Menni COVID-19 prediction model (i.e. predicted COVID-19 score = -1.32 –(0.01 * age) + (0.44 * male sex) + (1.75 * loss of smell and taste) + (0.31 * severe or significant persistent cough) + (0.49 * severe fatigue) + (0.39 * skipped meals)) and calculated the predicted probability of COVID-19 according to exp(predicted COVID-19 score)/(1+exp(predicted COVID-19 score)). The predictive properties of the model were tested using an ROC analysis by comparing the predicted probability of COVID-19 to the self-reported SARS-CoV-2 reverse-transcription PCR (RT-PCR) test outcome. Sensitivity, specificity, positive predictive value (PPV) and negative predictive value (NPV) were calculated based on a predicted probability higher than 0.50 to define a positive predicted case.

## Attempt to improve the Menni COVID-19 prediction model

The three cohorts with self-reported SARS-CoV-2 reverse-transcription PCR (RT-PCR) test outcomes available (Helix, Lifelines and NTR) were used in an attempt to improve the Menni COVID-19 prediction model. Self-reported symptoms that were present in all three cohorts were used. Symptoms were reported on a 5-point or 7-point Likert-scale (Lifelines COVID-19 and NTR cohorts) and binary scale (Helix cohort). To categorize all symptoms into a binary variable, we assessed the appropriate cut-off values in the Lifelines COVID-19 cohort for each self-reported symptom by performing a logistic regression on subjects with a positive test outcome (n = 56) compared to subjects with a negative test outcome (n = 586). In these models, each symptom was investigated separately by using two dummy variables indicating symptom severity (low and intermediate/high) with the reference being the absence of the symptom. If only intermediate/high symptom severity was significantly associated with a positive test, we used this value as cut-off. If both low and intermediate/high symptom severity were significant, we used low symptom severity as cut-off.

The symptoms selected for this model had to be present for the entire data-collection period in all three cohorts, resulting in the following symptoms being selected: coughing-any, diarrhea/stomach ache, difficulty breathing, fever, loss of smell/taste, runny nose, sore throat and tired-any. Subsequently, we performed forward and backward stepwise logistic regression in the Lifelines COVID-19 Cohort to construct the model most predictive for a positive test outcome (p-in = 0.10 and p-out = 0.10). The predictive properties were tested using an ROC analysis. Sensitivity, specificity, PPV and NPV were calculated based on the predicted probability, favouring an optimal PPV. We then applied this model to the Helix and NTR cohorts and tested the predictive properties.

## Genome-wide association analysis

To investigate whether the predicted COVID-19 phenotype can help accelerate the search for host genetic factors that contribute to the susceptibility of developing COVID-19 symptoms, we performed a GWAS for predicted COVID-19 case-control status as part of the COVID-19 Host Genetics Initiative (C19HG) with a total of 1865 cases and 29174 controls (https://www.covid19hg.org/results/, release 3). All cohorts consist of individuals of European ancestry. See S2 Table for the full phenotype description and detailed analysis plan. Additional details on cohort level GWAS and C19HG meta-analysis are provided in the S1 Methods.

## Processing of GWAS results

We downloaded the third release of the results of the predicted COVID-19 meta-analysis (*Predicted COVID-19 from self-reported symptoms vs. predicted or self-reported non-COVID-19*) from the C19HG website (https://www.covid19hg.org/results/) (genome assembly GRCh37, retrieved on 02-07-2020). For a comparison with other COVID-19 phenotypes we downloaded meta-analyses for *hospitalized COVID-19 vs. population*, *COVID-19 vs. self-reported negative*, and *COVID-19 vs. population as well*. For these downloaded summary statistics, we added RSIDs where both the genomic location and alleles matched to a variant from dbSNP [14]. Variants were filtered on MAF > 0.01 (based on the aggregated allele frequency over all cohorts), after which we performed p-value informed LD pruning, also called clumping, using PLINK (v1.90b6.10 64-bit) [15, 16] and the European population from the 1000 Genomes Project (phase 3) as a reference panel [17]. For clumping, thresholds on the linkage disequilibrium and genomic distance were set to an $R^2$ of 0.2 and a distance of 250 kb respectively. In GWASs other than the predicted COVID-19 analysis, the maximum p-value of index variants was set to $5 \times 10^{-8}$. All other parameters were left as their default values.

## Comparison between predicted COVID-19 and three other GWASs

For the predicted COVID-19 GWAS, the top 20 independent SNPs were selected, and these SNPs were compared with their respective effects in other COVID-19 GWASs to determine if their effects replicated. The same was done using the independent genome-wide significant hits from each of the other COVID-19 GWASs.

## Comparison of COVID-19 GWASs with previously reported associations in viral infection phenotypes

First, genome-wide significant variants ($P \leq 5 \times 10^{-8}$) were selected for eight viral infection phenotypes from the NHGRI-EBI GWAS Catalog (accessed July 7, 2020) [18]. Next, the individual SNPs corresponding to each association were queried from the processed COVID-19 GWASs. For every viral infection SNP that we found in one of the four COVID-19 GWASs, we determined if the SNP replicated, dictated by the p-value of the association in the respective COVID-19 GWAS, the Bonferroni-corrected significance level calculated from the number of SNPs for a viral infection trait and an a priori Bonferroni adjusted alpha of 0.05.

To get a more concrete indication whether or not the COVID-19 GWASs showed an increased signal of previously reported viral infection associations, quantile-quantile plots were made and accompanying genomic inflation factors (λ) were calculated in the selection of SNPs that have previously been reported to be associated with the various viral infection traits. A significance value for every λ was calculated by simulating 1000 expected λ-values, calculating the consequent Z-score for the observed λ, and determining a two-tailed p-value. λ-values were simulated by sampling n values from a $\chi^2$-distribution (k = 1), where n corresponds to the number of p-values used to calculate the observed λ-value.

## Enrichment analysis

Within the predicted COVID-19 phenotype, we selected all variants with a p-value $\leq 5 \times 10^{-4}$ and used DEPICT [19] with default settings to search for enrichment in pathways and protein-protein interactions. We used a false discovery rate of 0.05.

**Table 1. Model diagnostics of the Menni COVID-19 prediction model in Helix, Lifelines and NTR.**

| Cohort | AUC (95% CI)[a] | Sensitivity | Specificity | Positive predictive value | Negative predictive value |
|---|---|---|---|---|---|
| Helix | 0.79 (0.725–0.869) | 0.481 | 0.905 | 0.419 | 0.924 |
| Lifelines | 0.824 (0.758–0.890) | 0.446 | 0.951 | 0.463 | 0.947 |
| NTR | 0.864 (0.822–0.905) | 0.415 | 0.936 | 0.596 | 0.876 |

[a]The model: -1.32–0.01*age + 0.44*male sex + 1.75*loss of smell or taste + 0.31*severe or significant persistent cough + 0.49*severe fatigue + 0.39*skipped meals. A predicted probability cut-off of > 0.50 is used to define a positive predicted case.

## Results

### Description of cohorts

The Generation Scotland, Helix, Lifelines and Netherlands Twin Register (NTR) cohorts include a total of 168 (0, 27, 56 and 85, respectively) positively tested COVID-19 cases and 1157 (0, 189, 586 and 382, respectively) negatively tested controls. Descriptive statistics of the cohorts are provided in S3 Table. Additional results comparing positive and negatively tested individuals in Lifelines are presented in the S1 Results.

### Replication of the Menni COVID-19 prediction model in Helix, Lifelines and NTR

Table 1 presents the model diagnostics of the replication of the Menni COVID-19 prediction model in the three independent cohorts. The Menni model yields an area under the curve (AUC) ranging between 0.79 and 0.86 across the three cohorts, similar to the performance reported in the original study. Associations between predicted COVID-19 and the presence of specific co-morbidities in Lifelines are presented in the S1 Results.

### A new Lifelines prediction model for COVID-19 yields similar performance

Using self-reported symptoms of 56 positive and 586 negative test outcome cases in the Lifelines cohort, we next attempted to improve on the Menni COVID-19 prediction model. The results of the logistic regression used to determine the optimal cut-off values to recode the symptoms into binary variables are presented in S4 Table. The best prediction model was: -4.497 + 1.032 × cough + 2.042 × fever + 2.145 × loss of smell or taste. The estimates of the model and the predicted probability cut-off used to define a positive predicted case are presented in S5a and S5b Table. S5c Table shows the diagnostics of this Lifelines model in all 3 cohorts. Overall, the prediction accuracies of the two models are comparable. As the Menni COVID-19 prediction model was developed and validated in two larger cohorts, we decided to continue with case prediction based on the Menni COVID-19 prediction model in the subsequent GWAS.

### The first GWAS of predicted COVID-19

We conducted a GWAS meta-analysis on 1,865 predicted cases and 29,174 controls across four independent cohorts. The full summary statistics of our analysis are available for download online on the C19HG website [8]. The results of the top 20 ($P < 5.1 \times 10^{-6}$) independent single nucleotide polymorphisms (SNPs) for predicted COVID-19 are shown in Fig 2. Suggestive evidence of association with predicted COVID-19 was found for two SNPs (rs11844522, p = $1.9 \times 10^{-7}$; rs5798227, p = $2.2 \times 10^{-7}$) (S1 Fig).

**Comparison between the predicted COVID-19 phenotype and three other GWASs from the COVID-19 Host Genetics Initiative**

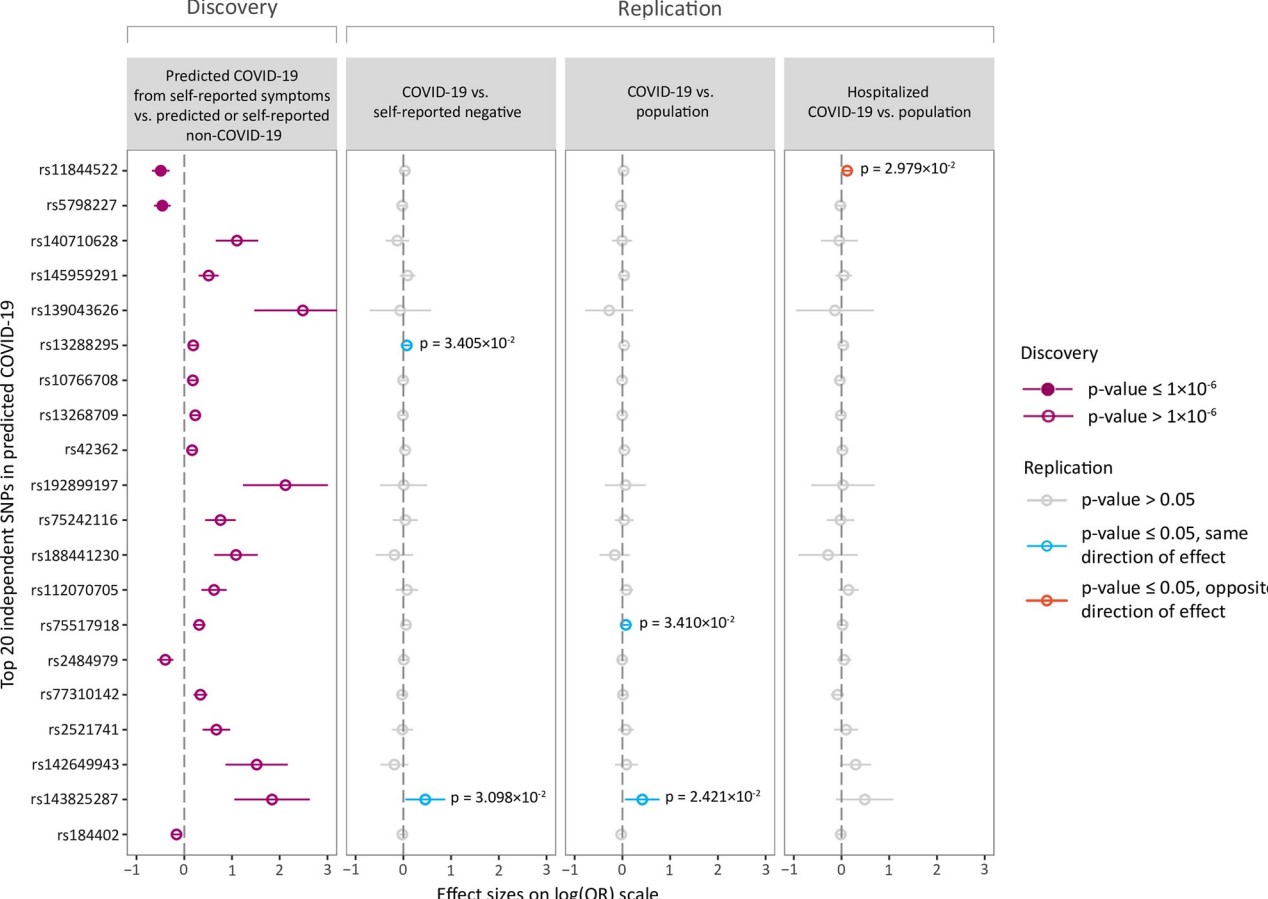

**Fig 2. Overview of the top loci associated with predicted COVID-19.** Shown are the effect size estimates of the top 20 independent SNPs associated with predicted COVID-19 and each of their associations with COVID-19 vs. self-reported negative, COVID-19 vs. population and Hospitalized COVID-19 vs. population. The effect sizes are shown with the risk allele odds ratio (OR) on a log-scale with a corresponding 95% confidence interval (CI). Colours indicate various p-value thresholds as described in the figure legend.

A comparison of the top 20 SNPs for predicted COVID-19 (*Predicted COVID-19 from self-reported symptoms vs. predicted or self-reported non-COVID-19*) with three other COVID-19 phenotypes showed three SNPs to be associated with the same direction of effect (rs13288295 in *COVID-19 vs. self-reported negative*, rs75517918 in *COVID-19 vs. population* and rs143825287 in both of these two phenotypes) and one SNP to be associated with the opposite direction of effect (rs11844522 in *Hospitalized COVID-19 vs. population*), based on a p-value threshold of 0.05 (Fig 2).

The meta-analyses of the *COVID-19 vs. population* GWAS showed an independent genome-wide significant association on locus 3p21.31 (rs35652899, p = 9.5x10[-11]). The *hospitalized COVID-19 vs. population* GWAS also showed two approximately independent genome-wide significant associations, of which the most significant is in high linkage disequilibrium with the associated variant from the *COVID-19 vs. population* analysis (rs35044562, p = 3.1x10[-15], $R^2$ = 0.97). A comparison of these results to the *predicted COVID-19* GWAS showed that both these associations with closely linked variants did not replicate at a significance level of 0.05 (p-values 0.18 and 0.22, respectively).

Next, we examined whether previously reported genetic associations with common viral infections share any overlap with the variants identified by our GWAS on COVID-19 susceptibility. After querying the NHGRI-EBI GWAS Catalog, we further investigated 270 genome-wide significant SNPs associated with known viral infections. Here we observe no major overlap with predicted COVID-19 at the level of individual SNPs (a priori Bonferroni-adjusted alpha = 0.05, S2 Fig). Out of 270 tested genome-wide significant SNPs, five replicate in one of the assessed COVID-19 phenotypes. Of these five variants, we found only a single variant to replicate in the predicted COVID-19 phenotype (rs3806400, $p = 3.077 \times 10^{-4}$). Furthermore, there was no overall increase in genomic inflation ($\lambda$) when considering all 270 SNPs jointly for any of the four GWAS phenotypes ($\lambda = 0.815$, $p = 0.2$ for *predicted COVID-19*, $\lambda = 1.234$, $p = 0.1$ for COVID-19 vs. self-reported negative, $\lambda = 1.110$, $p = 0.5$ for COVID-19 vs. population, $\lambda = 0.780$, $p = 0.1$ for hospitalized COVID-19 vs. population, respectively) (S3 Fig).

Downstream analysis using DEPICT [19] ascertained that protein-protein interactions with the solute carrier family 25 member 6 gene (*SLC25A6)* are significantly enriched ($p = 8.6 \times 10^{-6}$). Full results are provided in S6 Table.

## Discussion

We investigated if symptom-based prediction of potential COVID-19 cases can aid in the search for host genetic factors that contribute to COVID-19 susceptibility. We confirm that self-reported disease-related symptoms are useful for the prediction of infection status and report the first genome-wide association analysis on predicted COVID-19 in the C19HG consortium. We find suggestive evidence for rs11844522 and rs5798227 to be associated with predicted COVID-19 and observe no evidence for overlap with known genetic associations with other viral infections.

While the Menni prediction model has reasonable predictive properties, with AUCs ranging between 0.74 and 0.86 in the included cohorts, it yielded lower sensitivities (0.42 to 0.48) and positive predictive values (0.42 to 0.60). This indicates that a significant proportion of COVID-19 cases will be missed by the prediction model (false negatives) and that positive predicted cases will include false positives. As our attempt to improve this model did not result in improved predictions, this remains an avenue to explore for future work. Symptom prevalence before and after testing in Lifelines suggests that repeated self-report assessments of disease-related symptoms may offer finer resolution to further increase prediction accuracy.

The predicted COVID-19 phenotype can help increase the number of cases for genetic analyses of COVID-19. While GWASs can benefit from larger sample sizes, caution should be taken when applying more loose phenotyping, as such an approach can produce a smaller and less-specific genetic signal [20]. The predicted positive COVID-19 cases could include false positive cases that have underlying conditions, such as other viral infections or common diseases, that share symptomatology with the symptoms included in the prediction model. This may have subsequently confounded our GWAS, yielding results that are less specific for COVID-19 and more related to genetic susceptibility to general immune defence or potentially even conditions un-related to COVID-19. Out of the 270 genome-wide significant variants associated with other infectious diseases, only one variant replicated in the predicted COVID-19 phenotype (rs3806400, $p = 3.077 \times 10^{-4}$). Additionally, the calculated genomic inflation factors showed no inflation for viral infection SNPs across any of the four COVID-19 phenotypes. Based on these first results, we conclude there is minimal overlap between the genetic predisposition of COVID-19 and other infectious diseases.

The outcomes of the GWAS meta-analysis of predicted COVID-19 showed suggestive association for the SNPs rs11844522 and rs5798227. Interestingly, rs11844522 is in a locus

comprising immunoglobulins, and the closest mapping gene (*IGHV3-7*) to rs11844522 is part of a gene family that is enriched in total VDJ expression of COVID-19 patients in single-cell transcriptomic data [21]. Rs11844522 replicated with an opposite direction of effect in the *hospitalized COVID-19 vs. population* phenotype, and this is likely explained by the difference in phenotype of that GWAS meta-analysis, which focused on COVID-19 severity (i.e. susceptibility to a poor outcome) rather than COVID-19 susceptibility (i.e. susceptibility to developing COVID-19 symptoms). The lack of genome-wide significant SNPs in our predicted COVID-19 GWAS could be due to the smaller sample size. Among the 20 most-significant top-variants, we observe three variants with a nominal significant genetic association signal (at $p < 0.05$) in the meta-analyses on the *COVID-19 vs. population* or the *COVID-19 vs. self-reported negative* phenotypes, all with the same direction of effect.

A comparison between COVID-19 GWASs showed that the GWAS on predicted COVID-19 was unable to replicate the top genome-wide significant hit of the GWAS on positively tested COVID-19 cases vs. the general population (*COVID-19 vs. population*). Interestingly, a closely linked variant in the analysis that considers COVID-19 severity (*hospitalized COVID-19 vs. population*), is even more significant than the top variant in *COVID-19 vs. population* (rs35044562: $p = 3.1 \times 10^{-15}$ and rs35652899: $p = 8.6 \times 10^{-10}$, respectively, $R^2 = 0.97$). In the latter analysis, only the COVID19-Host(a)ge cohort, which contributed the largest number of cases, showed a genome-wide significant association with the top variant at this locus ($p = 8.2 \times 10^{-11}$), while other cohorts all showed much less significance (UK Biobank: $p = 1.3 \times 10^{-3}$; other cohorts: $p > 0.01$). Looking into this discrepancy revealed that the COVID19-Host(a)ge cohort focusses on severe COVID-19 patients, which is indicated by the fact that the cases contributed by this cohort to the *COVID-19 vs. population* GWAS completely overlap with hospitalized cases contributed to the GWAS on COVID-19 severity. A similar observation can be made for the UK Biobank, for which the number of hospitalized cases contributed to the *hospitalized COVID-19 vs. population* analysis constitute 66% of cases in *COVID-19 vs. population*, an observation that explains the increased significance for the association in this cohort compared to others. Taking these observations into account, it seems reasonable to assume that the reported variants on the 3p21.31 locus are more likely to be associated with COVID-19 severity than COVID-19 susceptibility. Therefore, no conclusion can be made on the performance of the predicted COVID-19 phenotype as a proxy for COVID-19 susceptibility based solely on the absence of an association with the 3p21.31 locus.

Downstream DEPICT analysis of the GWAS outcomes identified a significant enrichment of protein-protein interactions with *SLC25A6*. This gene encodes adenine nucleotide translocator 3 (ANT3), which is a core component of the mitochondrial permeability transition pore (MPTP) and is involved in apoptosis. *SLC25A6* is downregulated in human cytomegalovirus infection and associated with influenza virus–induced apoptosis [22, 23]. This indicates this gene might also be relevant to COVID-19 susceptibility.

## Limitations

There are multiple limitations of using predicted COVID-19 cases that we need to consider. Firstly, the training data might not be fully representative of the whole spectrum of COVID-19 symptoms since testing of putative cases in the early months of the pandemic was mostly restricted to patients with a more severe phenotype. Individuals with essential occupations, for example healthcare professionals, were also more frequently tested at the beginning of the pandemic. Secondly, some symptoms are also present in common chronic diseases, for example "loss of smell and taste" is frequent among patients with a neurological disorder. Indeed, a preliminary analysis of the Lifelines data showed enrichment of patients with pre-existing

conditions in the predicted COVID-19 cases as compared to controls but no enrichment in the confirmed COVID-19 cases compared to confirmed negative cases, indicating that these individuals might be incorrectly predicted as COVID-19 cases by the Menni COVID-19 prediction model based on their symptoms (S4 Fig). Thirdly, the prevalence of COVID-19 might be different among different populations and cohorts. The false positive rates of the prediction models are likely to be larger if the prevalence of COVID-19 is small compared to other infectious diseases that often have similar symptoms.

## Conclusions

In an effort to identify host genetic factors that contribute to the susceptibility of COVID-19, we have conducted a GWAS on symptom-based prediction of COVID-19. While we demonstrated that the Menni et al. COVID-19 prediction model has reasonable and consistent performance across multiple independent cohorts, the GWAS on the predicted phenotype did not yield genome-wide significant loci. Explorative analyses of the genetic overlap between predicted COVID-19 and other viral infectious diseases, suggest that genetic variants involved in other viral infectious diseases do not overlap with COVID-19 susceptibility and that COVID-19 severity may have a partially different underlying genetic architecture. Our study shows that the inclusion of symptom-based predicted cases could be a useful strategy in a scenario of limited testing, either during the current COVID-19 pandemic or any future viral outbreak.

## Supporting information

**S1 Fig. Regional association plots for the locus of the two top SNPs rs11844522 (A) and rs5798227 (B) in the predicted COVID-19 GWAS (p $\leq$ 1x10$^{-6}$).** In each of the two panels, the top SNP is indicated by a purple diamond. Other SNPs are colored according to their linkage disequilibrium with the top SNP (calculated with the European population from the 1000 Genomes Project (phase 3) as a reference panel). The genes located within the visualized regions are drawn at their respective locations, with an arrow indicating the transcribed strand. Positions correspond to genome assembly GRCh37.
(PDF)

**S2 Fig. Comparison of genome-wide significant independent SNPs discovered in other viral infection phenotypes replicated in the four COVID-19 phenotypes Hospitalized COVID-19 vs. population, COVID-19 vs. self-reported negative, COVID-19 vs. population, predicted COVID-19 from self-reported symptoms vs. predicted or self-reported non-COVID-19.** In each of the 5 GWASs in which we replicated an association in a COVID-19 phenotype Europeans constitute the largest portion of included samples. Three of these contained only European samples. The majority of the samples in the COVID-19 analyses are also of European descent. This suggests that these lead SNPs can be compared, and that replication was not caused by ethnicity differences.
(PDF)

**S3 Fig. Quantile-quantile plots (Q-Q plots) of the four phenotypes: Hospitalized COVID-19 vs. population, COVID-19 vs. self-reported negative, COVID-19 vs. population and predicted COVID-19 from self-reported symptoms vs. predicted or self-reported non-COVID-19, wherein variants are confined to a selection of associated with several viral infection traits.** For every COVID-19 GWAS, a genomic inflation factor ($\lambda$) with accompanying p-value is shown. Based on both these values and the Q-Q plots, there is no indication of

an increased viral infection signal in one of the COVID-19 phenotypes.
(PDF)

**S4 Fig. Chronic disease associations with COVID-19 predicted cases and COVID-19 positive cases.** Fisher's exact test shows a Bonferroni significant positive correlation between "Neurological disease", "Psychological disease", "Chronic muscle disease", "Cancer" and "Lung disease" patients and COVID-19 predicted cases. This association is not present for positive COVID-19 cases. Applying generalised linear models with "age", "sex" and "bmi" as covariates, instead of Fishers exact tests, does not change which diseases are Bonferroni significant and which are not.
(PDF)

**S1 Table. The different cut-offs and symptoms that were used by the Generation Scotland, Helix, Lifelines and NTR cohorts when applying the Menni COVID-19 prediction model to the datasets prior to running the GWAS.**
(DOCX)

**S2 Table. Full phenotype description and detailed analysis plan used within the C19HG for genome-wide association analysis.**
(DOCX)

**S3 Table. Descriptive statistics of the Generation Scotland, Helix, Lifelines and NTR cohorts.** Due to absence of testing data, Generation Scotland could not be used for replication of the Menni COVID-19 prediction model and in the development of the Lifelines COVID-19 prediction model.
(DOCX)

**S4 Table. Overlapping symptoms in the Helix, Lifelines and NTR cohorts.** Logistic regression for each symptom separately in Lifelines on positive (n = 56) vs negative (n = 586) tested subjects to define symptom cut-offs (reference = absence of symptom).
(DOCX)

**S5 Table.** The Lifelines COVID-19 prediction model (a). Diagnostics of different cut-offs of predicted probability of the Lifelines COVID-19 prediction model (b). Model diagnostics of the Lifelines COVID-19 prediction model in the Helix, Lifelines and NTR cohorts (c).
(DOCX)

**S6 Table. Downstream analysis of DEPICT.**
(XLSX)

**S1 Results. Prevalence of core predicted COVID-19 symptoms in Lifelines and phenotypic associations between predicted COVID-19 and co-morbidities in Lifelines.**
(DOCX)

**S1 Methods. Detailed information on cohort level GWAS and C19HG meta-analysis.**
(DOCX)

## Acknowledgments

We want to thank all the study participants that have donated—and still are donating—samples to help research on COVID-19. The COVID-19 Host Genetics Initiative was originally initiated by Andrea Ganna and Mark Daly, but it belongs to all the participating cohorts.

We thank the UMCG Genomics Coordination Center, the UG Center for Information Technology and their sponsors BBMRI-NL & TarGet for storage and compute infrastructure. We thank Kate Mc Intyre for editing this manuscript.

## Consortia section

### Lifelines COVID-19 Cohort authors

HM Boezen (1), Jochen O. Mierau (2,3), H. Lude Franke (4), Jackie Dekens (4,6), Patrick Deelen (4), Pauline Lanting (4), Judith M. Vonk (1), Ilja Nolte (1), Anil P.S. Ori (4,5), Annique Claringbould (4), Floranne Boulogne (4), Marjolein X.L. Dijkema (4), Henry H. Wiersma (4), Robert Warmerdam (4), Soesma A. Jankipersadsing (4), Irene van Blokland (4,7).

1. Department of Epidemiology, University of Groningen, University Medical Center Groningen, Groningen, The Netherlands

2. Faculty of Economics and Business, University of Groningen, Groningen, The Netherlands

3. Aletta Jacobs School of Public Health, Groningen, The Netherlands

4. Department of Genetics, University of Groningen, University Medical Center Groningen, Groningen, The Netherlands

5. Department of Psychiatry, University of Groningen, University Medical Center Groningen, Groningen, The Netherlands

6. Center of Development and Innovation, University of Groningen, University Medical Center Groningen, Groningen, The Netherlands

7. Department of Cardiology, University of Groningen, University Medical Center Groningen, Groningen, The Netherlands

### The COVID-19 Host Genetics Initiative authors

1. **BioMe**
   **Data collection and coordination**: Judy H. Cho, Ruth J.F. Loos
   **Analysis**: Arden Moscati

2. **Corea (Genetics of COVID-related Manifestation)**
   **Data collection and coordination**: Kangbuk Samsung Cohort Study (KSCS), Yoosoo Chang, Pyoeng Gyun Choe, Jin Chung, Sinyoung Ham, Eun-Jeong Joo, Jongtak Jung, Chang Kyung Kang, Hyung-Lae Kim, Hong Bin Kim, Eu Suk Kim, Hyo-Jung Lee, Sookyung Park, Kyoung-Un Park, Jeong Su Park, Seungho Ryu, Kyoung-Ho Song
   **Technical support**: Global Science Experimental Data Hub Center (GSDC), Korea Research Environment Open Network (KREONET)
   **Analysis**: Han-Na Kim
   **Administrative support**: Nam-Jong Paik

3. **COVID19-Host(a)ge**
   **Ethics and communication**: Agustín Albillos, Rosanna Asselta, Luis Bujanda, Maria Buti, Stefano Duga, Javier Fernández, Manuel Romero Gomez, Pietro Invernizzi, Daniele Prati
   **Data collection and coordination**: Jesus M. Banales, Trine Folseraas, Andre Franke,

Johannes R Hov, Tom H Karlsen, Luca Valenti
**Analysis**: Frauke Degenhardt, David Ellighaus

4. **deCODE genetics**
**Data collection and coordination**: Elias S Eythorsson, Asgeir Haraldsson, Dadi Helgason, Hilma Holm, Ragnar F Ingvarsson, Ingileif Jonsdottir, Gudmundur L Norddahl, Runolfur Palsson, Jona Saemundsdottir, Kari Stefansson, Unnur Thorsteinsdottir
**Analysis**: Daníel F Gudbjartsson, Hakon Jonsson, Pall Melsted, Patrick Sulem, Gardar Sveinbjornsson

5. **Determining the Molecular Pathways and Genetic Predisposition of the Acute Inflammatory Process Caused by SARS-CoV-2**
**Data collection and coordination**: Marta E. Alarcón-Riquelme, David Bernardo, Silvia Rojo Rello
**Analysis**: Manuel Martínez-Bueno

6. **Finngen**
**Data collection and coordination**: Finngen

7. **GEN-COVID, reCOVID**
**Technical support**: Sara Amtrano, Mirella Bruttini, Valentino Floriana, Anna Rita Giliberti
Analysis: Elisa Benetti, Chiara Fallerini, Simone Furini, Anna Maria Pinto
**Data collection and coordination**: Margherita Baldassarri, Francesca Fava, Francesca Mari, Alessandra Renieri
**Other**: Susanna Croci, Rossella Tita
**Ethics and communication**: Elisa Frullanti

8. **Generation Scotland**
**Analysis**: Drew Altshul, Archie Campbell, Caroline Hayward
**Data collection and coordination**: Chloe Fawns-Ritchie, David Porteous

9. **Genes & Health**
Data collection and coordination: Qinqin Huang, Karen A Hunt, Hilary C Martin, Dan Mason, Richard C Trembath, Bhavi Trivedi, John Wright
**Other**: Sarah Finer, Christopher Griffiths
**Analysis**: David A van Heel

10. **Genetic determinants of COVID-19 complications in the Brazilian population**
**Data collection and coordination**: Cinthia E Jannes, Jose E Krieger, Alexandre C Pereira
**Technical support**: Emmanuelle Marques

11. **Genomic epidemiology of SARS-Cov-2 and host genetics in Coronavirus Disease 2019 (COVID-19)**
**Technical support**: Karen Dalton, Christopher DeBoever, Illumina, Inc., David Jimenez-Morales, Aldo Cordova Palomera, Benjamin Pinsky, Erin Smith, Sandor Szalma, Cathy Tralau-Stewart, Emily Wong
**Data collection and coordination**: John Gorzynski, Hannah de Jong
**Analysis**: David Amar, Olivier Delaneau, Christopher Hughes, Alexander Ioannidis, Archana Raja, Simone Rubinacci, Yosuke Tanigawa
**Administrative support**: Euan Ashley, Carlos Bustamante, Vicki Parikh, Manuel Rivas, Matthew Wheeler

12. **Genetic modifiers for COVID-19 related illness**
**Data collection and coordination**: Adeline Busson, Jean-Christophe Goffard, Isabelle Migeotte, Xavier Peyrassol, Guillaume Smits, Isabelle Vandernoot, Francoise Wilkin
**Technical support**: Youssef Bouysran, Bruno Pichon, Nicky Tiembe

13. **Helix Exome+ COVID-19 Phenotypes**
Data collection and coordination: Kelly M. Schiabor Barrett, Alexandre Bolze, Elizabeth T. Cirulli, Jimmy M. Ramirez III, Yan Wei Lim, James T. Lu, Stephen Riffle, Francisco Tanudjaja, Xueqing Wang, Nicole L. Washington, Simon White

14. **Lifelines**
**Analysis**: Annique Claringbould, Patrick Deelen, Esteban Lopera, Robert Warmerdam
**Data collection and coordination**: Marike Boezen, Lude Franke

15. **Mass General Brigham–Host Vulnerability to COVID-19**
**Data collection and coordination**: Robert Green, Beth Karlson, James Meigs, Josep Mercader, Shawn Murphy, Emma Perez, Sue Slaugenhaupt, Jordan Smoller, Scott Weiss, Ann Woolley
**Analysis**: Yen-Chen Anne Feng

16. **Netherlands Twin Register**
**Data collection and coordination**: Meike Bartels, Eco de Geus, Michel G Nivard
**Analysis**: Jouke-Jan Hottenga

17. **Population controls**
**Data collection and coordination**: Alfredo Brusco, Cynthia M Bulik, Denis Franchimont, Mikael Landen, Edouard Louis, Nancy Pedersen, Souad Rahmouni, Pasquale Striano, Severine Vermeire, Federico Zara

18. **Qatar Genome Program**
**Data collection and coordination**: Wadha Al-Muftah, Radja Badji, Said Ismail
**Analysis**: Yasser Al-Sarraj, Hamdi Mbarek

19. **SIGMA**
**Data collection and coordination**: Marta E. Alarcón-Riquelme

20. **UK 100,000 Genomes Project**
**Data collection and coordination**: Prabhu Arumugam, Mark Caulfield, Genomics England Research Consortium, Anna C Need, Thomas Oscroft, Augusto Rendon, Richard H Scott
**Analysis**: Georgia Chan, Athanasios Kousathanas, Loukas Moutsianas, Chris A Odhams, Dorota Pasko, Dan Rhodes, Alex Stuckey

21. **UK Biobank**
**Analysis**: Elizabeth G. Atkinson, Nikolas Baya, Guillaume Butler-Laporte, Hilary Finucane, Vincenzo Forgetta, Masahiro Kanai, Konrad J. Karczewski, Nils Koelling, Alicia R. Martin, Tomoko Nakanishi, Duncan S. Palmer, J. Brent Richards, Chris C A Spencer, Patrick Turley, Raymond K. Walters, Daniel J Wilson
**Data collection and coordination**: Jacob Armstrong, Anne Marie O'Connell, David H Wyllie
**Technical support**: Sam Bryant
**Administrative support**: Claire Churchhouse

22. **UK Blood Donors Cohort**
   **Data collection and coordination**: Emanuele Di Angelantonio, Michael Chapman, John Danesh, Willem Ouwehand, Dave Roberts, Nick Watkins
   **Analysis**: Adam Butterworth, Jing Hua Zhao

23. **COVID-19 Host Genetics Initiative Coordination**
   **Phenotype steering group**: Les Biesecker, Lea Davis, Patrick Deelen, Andrea Ganna, David van Heel, Eric Kerchberger, Sulggi Lee, Tomoko Nakanishi, James Priest, Alessandra Renieri, Brent Richards, Vijay Sankaran
   **Administrative support**: Karolina Chwialkowska, Margherita Francescatto, Christine Stevens
   **International Common Disease Alliance**: Amy Trankiem, Kate Balaconis
   **Leadership**: Rachel Liao, Mark Daly, Andrea Ganna, Ben Neale
   Data dictionary: Anna Bernasconi, Stefano Ceri, Francesca Mari, Alessandra Renieri
   **Analysis**: Juha Karjalainen, Mattia Cordioli, Mari Niemi, Wei Zhou
   **Website**: Huy Nguyen, Matthew Solomonson

24. **In silico follow-up results**
   **Analysis**: Hilary Finucane, Shyamalika Gopalan, Kangcheng Hou, Philip Jansen, Masahiro Kanai, Christiaan de Leeuw, Zeyun Lu, Nicholas Mancuso, Eirini Marouli, Areti Papadopoulou, Bogdan Pasaniuc, Gita Pathak, Renato Polimanti, Danielle Posthuma, Jeanne Savage, Emil Uffelmann, Peter Visscher, Frank R Wendt, Naomi Wray, Loic Yengo

## Author Contributions

**Conceptualization:** Irene V. van Blokland, Pauline Lanting, Anil P. S. Ori, Judith M. Vonk, Robert C. A. Warmerdam, Johanna C. Herkert, Floranne Boulogne, Elizabeth T. Cirulli, Eco J. C. de Geus, Patrick Deelen, Lude H. Franke.

**Data curation:** Irene V. van Blokland, Pauline Lanting, Anil P. S. Ori, Judith M. Vonk, Robert C. A. Warmerdam, Johanna C. Herkert, Meike Bartels, Jouke-Jan Hottenga, Andrea Ganna, Juha Karjalainen, Caroline Hayward, Chloe Fawns-Ritchie, Archie Campbell, David Porteous, Elizabeth T. Cirulli, Kelly M. Schiabor Barrett, Stephen Riffle, Alexandre Bolze, Simon White, Francisco Tanudjaja, Xueqing Wang, Yan Wei Lim, James T. Lu, Nicole L. Washington, Eco J. C. de Geus, Patrick Deelen, Lude H. Franke.

**Formal analysis:** Anil P. S. Ori, Judith M. Vonk, Robert C. A. Warmerdam, Floranne Boulogne, Annique Claringbould, Esteban A. Lopera-Maya, Caroline Hayward, Chloe Fawns-Ritchie, Archie Campbell, Eco J. C. de Geus, Patrick Deelen.

**Funding acquisition:** Lude H. Franke.

**Investigation:** Irene V. van Blokland, Pauline Lanting, Anil P. S. Ori, Judith M. Vonk, Robert C. A. Warmerdam, Meike Bartels, Jouke-Jan Hottenga, Andrea Ganna, Juha Karjalainen, Caroline Hayward, Chloe Fawns-Ritchie, Archie Campbell, David Porteous, Kelly M. Schiabor Barrett, Stephen Riffle, Alexandre Bolze, Simon White, Francisco Tanudjaja, Xueqing Wang, Yan Wei Lim, James T. Lu, Nicole L. Washington, Eco J. C. de Geus, Patrick Deelen, Lude H. Franke.

**Methodology:** Irene V. van Blokland, Pauline Lanting, Anil P. S. Ori, Judith M. Vonk, Robert C. A. Warmerdam, Johanna C. Herkert, Floranne Boulogne, Elizabeth T. Cirulli, Eco J. C. de Geus, Lude H. Franke.

**Project administration:** Pauline Lanting, Lude H. Franke.

**Resources:** Elizabeth T. Cirulli, Jimmy M. Ramirez, III.

**Supervision:** H. Marike Boezen, Lude H. Franke.

**Validation:** Judith M. Vonk.

**Visualization:** Irene V. van Blokland, Pauline Lanting, Anil P. S. Ori, Judith M. Vonk, Robert C. A. Warmerdam, Floranne Boulogne, Patrick Deelen, Lude H. Franke.

**Writing – original draft:** Irene V. van Blokland, Pauline Lanting, Anil P. S. Ori, Judith M. Vonk, Robert C. A. Warmerdam, Elizabeth T. Cirulli, Patrick Deelen, Lude H. Franke.

**Writing – review & editing:** Irene V. van Blokland, Pauline Lanting, Anil P. S. Ori, Judith M. Vonk, Robert C. A. Warmerdam, Johanna C. Herkert, Floranne Boulogne, Caroline Hayward, Eco J. C. de Geus, Patrick Deelen, Lude H. Franke.

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
