## [Decision Letter · Decision Letter 0]

15 Feb 2021

PONE-D-20-38159

Using symptom-based case predictions to identify host genetic factors that contribute to COVID-19 susceptibility

PLOS ONE

Dear Dr. van Blokland,

Thank you for submitting your manuscript to PLOS ONE. After careful consideration, we feel that it has merit but does not fully meet PLOS ONE’s publication criteria as it currently stands. Therefore, we invite you to submit a revised version of the manuscript that addresses the points raised during the review process.

We look forward to receiving your revised manuscript.

Kind regards,

Wenbin Tan

Academic Editor

PLOS ONE

2.We note that the grant information you provided in the ‘Funding Information’ and ‘Financial Disclosure’ sections do not match.

3.We note that you have indicated that data from this study are available upon request. PLOS only allows data to be available upon request if there are legal or ethical restrictions on sharing data publicly. For information on unacceptable data access restrictions, please see http://journals.plos.org/plosone/s/data-availability#loc-unacceptable-data-access-restrictions.

5. One of the noted authors is a group or consortium (The COVID-19 Host Genetics Initiative,). In addition to naming the author group, please list the individual authors and affiliations within this group in the acknowledgments section of your manuscript. Please also indicate clearly a lead author for this group along with a contact email address.

7.Thank you for stating the following in the Competing Interests section:

"ETC, KMSB, SR, AB, SW, FT, XW, JMR, YWL, JTL, and NLW are employees of Helix. All other authors declare no financial or non-financial conflict of interest."

We note that one or more of the authors are employed by a commercial company: Helix

Editor: The manuscript performed a GWAS analysis to show possible COVID-19 associated SNPs. However, the significance of the results was substantially undermined due to (1) the very limited confirmed positive cases of COVID-19 in the study (n=168), and (2) the low sensitivity of Mennis COVID-19 prediction model. Based on the current available data of COVID-19 cases, a substantial increase in confirmed COVID-19 cases in the study will help provide a more reliable conclusion.

Reviewer #1: The manuscript describes mainly negative results from a prediction model based on the symptoms to a GWAS analysis. The study makes almost no suggestive conclusions but is important to publish these results nonetheless. Most of my comments would rather focus on the "what I wish were done" rather than requirements for publication since given the methodology and results and the absence of any strong conclusions, the statements made are not unreasonable.

- I found the number of cases in this study too small. This is a limitation of the data, however, I think there could me more work done to increase the cohort given the widespread access to COVID-19 patient data.

- The racial diversity is quite lacking in the data to consider the effort done in GWAS analysis of high value.

- I do not understand why a conversion into binary variables was required for each of the variables in preprocessing. A trainable model is capable of dealing with ordinal variables without any conversion. Is it due to the heterogeneity between symptom reporting in different data sets?

The following comments are requirements for clarification of the methodology

- Logistic regression models are ML models, which means that there is a training phase. A complete description of the model hyperparameters and training procedures (training/test splits) is required to assess the validity of the model.

- There are multiple mathematical/statistical notions that need to be properly typed (italic p for p-value... etc.)

---

## [Author Response · Author response to Decision Letter 0]

13 Apr 2021

We are grateful for the useful feedback of reviewers and thank the editor for the opportunity to revise our manuscript. In a point-by-point manner, we answer the reviewer's comments in the "Response to Reviewers" document and hope that this addresses any remaining concerns for publication in PLOS ONE.

---

## [Decision Letter · Decision Letter 1]

3 Jun 2021

PONE-D-20-38159R1

Using symptom-based case predictions to identify host genetic factors that contribute to COVID-19 susceptibility

PLOS ONE

Dear Dr. van Blokland,

Thank you for submitting your manuscript to PLOS ONE. After careful consideration, we feel that it has merit but does not fully meet PLOS ONE’s publication criteria as it currently stands. Therefore, we invite you to submit a revised version of the manuscript that addresses the points raised during the review process.

We look forward to receiving your revised manuscript.

Kind regards,

Wenbin Tan

Academic Editor

PLOS ONE

Journal Requirements:

Review Comments to the Author

Reviewer #2: This manuscript reports symptom-based predictive phenotypes as proxies for COVID-19 to understand genetic susceptibility to COVID-19. The strategy is well thought out and there are no problems. However, it is not clear whether or not differences due to the genetic background of the population have been taken into account. Are the eight genome-wide significantly different variants for viral infections selected from the NHGRI-EBI GWAS Catalog from a valid population for this comparison? Are GWAS meta-analyses of cohorts containing different genetic backgrounds valid? Should the current findings be considered valid only for specific populations? For example, rs5798227 associated with COVID-19 seems to be present in most Asians. It would be better if there was mention of the differences in genetic background among populations and how this affects the prediction model.

---

## [Author Response · Author response to Decision Letter 1]

28 Jun 2021

PONE-D-20-38159 

Authors: van Blokland et al.

Response to reviewers

We thank the editors and and reviewers for taking the time to consider our manuscript. Please see below point-by-point answer to the comments raised by the academic editor and reviewer.

"Journal Requirements:

Please review your reference list to ensure that it is complete and correct. If you have cited papers that have been retracted, please include the rationale for doing so in the manuscript text, or remove these references and replace them with relevant current references. Any changes to the reference list should be mentioned in the rebuttal letter that accompanies your revised manuscript. If you need to cite a retracted article, indicate the article’s retracted status in the References list and also include a citation and full reference for the retraction notice."

We have updated reference 11 from the following

Mc Intyre K, Lanting P, Deelen P, Wiersma H, Vonk JM, Ori AP, et al. The Lifelines COVID-19 Cohort: a questionnaire-based study to investigate COVID-19 infection and its health and societal impacts in a Dutch population-based cohort. medRxiv. 2020 Jun 24;2020.06.19.20135426. 

To the updated reference:

Mc Intyre K, Lanting P, Deelen P, Wiersma HH, Vonk JM, et al. Lifelines COVID-19 cohort: investigating COVID-19 infection and its health and societal impacts in a Dutch population-based cohort. BMJ Open. 2021 Mar 17;11(3):e044474. doi: 10.1136/bmjopen-2020-044474. 

"Reviewer #2: This manuscript reports symptom-based predictive phenotypes as proxies for COVID-19 to understand genetic susceptibility to COVID-19. The strategy is well thought out and there are no problems. However, it is not clear whether or not differences due to the genetic background of the population have been taken into account. Are the eight genome-wide significantly different variants for viral infections selected from the NHGRI-EBI GWAS Catalog from a valid population for this comparison? Are GWAS meta-analyses of cohorts containing different genetic backgrounds valid? Should the current findings be considered valid only for specific populations? For example, rs5798227 associated with COVID-19 seems to be present in most Asians. It would be better if there was mention of the differences in genetic background among populations and how this affects the prediction model."

Thank you for your remark, this is a valid consideration with two potential implications. The first is that the GWASs on the other viral infections might have been performed on non-European populations. If that were to be the case, a direct comparison of lead or top variants might not be straightforward. To ensure that this is not the case for the overlapping variants between our predicted COVID19 GWAS and the other viral infections we double checked this. All these GWASes are primarily conducted on Europeans. The cohorts that we used are also primarily of European descent so these lead variants can be compared, and a meta-analysis could be conducted without special considerations. We have included this statement in the manuscript in the description of Supplementary Figure 2. 

Secondly, there could indeed be a difference in minor allele frequencies among different populations. This could indicate that some populations might be slightly more or less susceptible to an infection. However the association is still relevant to other populations, this is extensively discussed here: https://www.nature.com/articles/s41586-020-2818-3

---

## [Decision Letter · Decision Letter 2]

16 Jul 2021

Using symptom-based case predictions to identify host genetic factors that contribute to COVID-19 susceptibility

PONE-D-20-38159R2

Dear Dr. van Blokland,

We’re pleased to inform you that your manuscript has been judged scientifically suitable for publication and will be formally accepted for publication once it meets all outstanding technical requirements.

Kind regards,

Wenbin Tan

Academic Editor

PLOS ONE

---

## [Editor Report · Acceptance letter]

2 Aug 2021

PONE-D-20-38159R2 

Using symptom-based case predictions to identify host genetic factors that contribute to COVID-19 susceptibility 

Dear Dr. van Blokland:

I'm pleased to inform you that your manuscript has been deemed suitable for publication in PLOS ONE. Congratulations! Your manuscript is now with our production department. 

Kind regards, 

on behalf of

Dr. Wenbin Tan 

Academic Editor

PLOS ONE